# Spontaneous Overactivation of *Xenopus* Frog Eggs Triggers Necrotic Cell Death

**DOI:** 10.3390/ijms25105321

**Published:** 2024-05-13

**Authors:** Alexander A. Tokmakov, Ryuga Teranishi, Ken-Ichi Sato

**Affiliations:** 1Institute of Advanced Technology, Faculty of Biology-Oriented Science and Technology, KinDai University, 930 Nishimitani, Kinokawa City 649-6493, Japan; 2Faculty of Life Sciences, Kyoto Sangyo University, Kamigamo-Motoyama, Kita-ku, Kyoto 603-8555, Japan; i2287166@cc.kyoto-su.ac.jp

**Keywords:** *Xenopus laevis*, eggs, spontaneous activation, mechanical stress, necrosis

## Abstract

The excessive activation of frog eggs, referred to as overactivation, can be initiated by strong oxidative stress, leading to expedited calcium-dependent non-apoptotic cell death. Overactivation also occurs spontaneously, albeit at a low frequency, in natural populations of spawned frog eggs. Currently, the cytological and biochemical events of the spontaneous process have not been characterized. In the present study, we demonstrate that the spontaneous overactivation of *Xenopus* frog eggs, similarly to oxidative stress- and mechanical stress-induced overactivation, is characterized by the fast and irreversible contraction of the egg’s cortical layer, an increase in egg size, the depletion of intracellular ATP, a drastic increase in the intracellular ADP/ATP ratio, and the degradation of M phase-specific cyclin B2. These events manifest in eggs in the absence of caspase activation within one hour of triggering overactivation. Importantly, substantial amounts of ATP and ADP leak from the overactivated eggs, indicating that plasma membrane integrity is compromised in these cells. The rupture of the plasma membrane and acute depletion of intracellular ATP explicitly define necrotic cell death. Finally, we report that egg overactivation can occur in the frog’s genital tract. Our data suggest that mechanical stress may be a key factor promoting egg overactivation during oviposition in frogs.

## 1. Introduction

Mature ovulated frog eggs are arrested in the metaphase of the second meiotic division by the high activity of the key meiotic regulators, the maturation promoting factor (MPF), and the cytostatic factor (CSF) [1]. The meiotic metaphase arrest prevents cell cycle progression and parthenogenesis prior to fertilization. While awaiting fertilization, meiotically arrested eggs grow older and experience various injuries, resulting in the loss of their quality. The stress- and age-acquired damage leads to decreased rates of fertilization, polyspermy, parthenogenesis, and the abnormal development of embryos. Poor-quality oocytes and eggs are considered to be a main cause of infertility and abnormal embryo development in different animals, including mammals [2,3].

As a consequence of progressive quality worsening, ovulated frog and mammalian eggs can be successfully fertilized only within several hours to days following ovulation. It was reported that in the absence of fertilization, ovulated mammalian eggs gradually deteriorate in the process of postovulatory aging, undergo fragmentation, and, eventually, degrade by apoptosis [4,5]. Furthermore, unfertilized eggs of the African clawed frog, *Xenopus laevis*, which are widely used in cell cycle and reproductive studies due to their high biochemical and cytological tractability, spontaneously activate, exit the meiotic metaphase arrest, and degrade by a well-defined apoptotic process, both in external aquatic environments and in the genital tract, within 48 h after ovulation [6,7]. It is widely recognized that spontaneous activation, which promotes exit from the meiotic metaphase arrest, makes successful fertilization in different species impossible [8,9].

The intracellular pathways involved in the spontaneous activation of metaphase II-arrested eggs are poorly understood. It was suggested that this process might employ a calcium-dependent mechanism in mammalian eggs [8,10,11]. Artificial elevation of intracellular calcium is known to cause the parthenogenetic activation of eggs from different species. However, spontaneous activation may also engage calcium-independent mechanisms. It was found that aged mouse and pig eggs have decreased activities of major CSF and MPF components [12,13]. Also, it was demonstrated that apoptosis is triggered by the progressive inactivation of the MAP kinase in aging unfertilized sea urchin eggs [14,15]. Thus, the gradual decrease in the content and/or activity of the key meiotic regulators has been speculated to cause meiotic exit in the absence of intracellular calcium signaling [9].

Currently, the physiological inducers of spontaneous egg activation remain unidentified. It was suggested that oxidative stress might cause this process, leading to expedited aging and the deterioration of postovulatory oocytes [16]. Indeed, hydrogen peroxide was shown to trigger the Src kinase- and calcium-dependent activation of *Xenopus* eggs [17]. The study also reported that prolonged treatment with hydrogen peroxide leads to excessive egg activation, i.e., overactivation, which gives rise to a very distinctive and easily identified egg phenotype. Fast and irreversible cortical contraction, lipofuscin accumulation, the depletion of intracellular ATP, a decline in mitochondrial membrane potential, and a decrease in the content of soluble cytoplasmic protein were observed in frog eggs overactivated by strong oxidative stress. These intracellular events develop in the absence of caspase activation [18,19]. It was concluded that overactivated eggs die by a distinctive calcium-dependent and non-apoptotic mechanism [19].

Notably, overactivation can occur spontaneously in the absence of any activation stimuli, albeit at a low frequency, in natural populations of frog eggs [18]. Spontaneous overactivation is viewed as a pathological and uncontrollable process; however, its cytological and biochemical events have not been characterized. The present study focuses on the spontaneous overactivation of spawned, as well as in vitro-matured, unfertilized *Xenopus* eggs. We demonstrate here that the main cytological and biochemical events of spontaneous overactivation mirror those of oxidative stress-induced overactivation. We also report that egg overactivation can occur in the frog’s genital tract and further propose that mechanical stress may be a key factor causing egg overactivation during oviposition in frogs. In addition, an overactivation-induced loss of plasma membrane integrity indicates that overactivated eggs die by necrosis.

## 2. Results

### 2.1. Spontaneous Overactivation of Spawned and In Vitro-Matured Xenopus Eggs

As reported previously, the oxidative stress-induced overactivation of *Xenopus* eggs is characterized by the fast and irreversible contraction of the egg’s cortical layer, an increase in egg size, and ATP depletion [18,19]. However, overactivation was induced by unphysiologically high concentrations of hydrogen peroxide in previous studies. Thus, we investigated whether overactivation in the absence of any activating stimuli would cause similar changes in the eggs.

It was found that the features of spontaneous overactivation, which occurred quite rarely in the populations of spawned *Xenopus* eggs, mirrored those of oxidative stress-induced overactivation. Cortical contraction proceeded very fast in spontaneously overactivated eggs, and within just several minutes, these cells whitened, presenting a very distinctive and easily recognizable phenotype (Figure 1A). Also, a significant increase in cell size, as well as ATP depletion, were observed in spontaneously overactivated spawned eggs within one hour (Figure 1B,C). In addition, similar changes were witnessed in defolliculated in vitro-matured eggs after their spontaneous overactivation (Figure 2). Notably, the morphology of apoptotic eggs differed significantly from that of overactivated eggs (Figure 2A). An increase in cell size and a decrease in intracellular ATP were also evident in late apoptosis; however, the changes were not as prominent as those observed in overactivated eggs (Figure 2B,C). Markedly, in contrast to apoptotic eggs, no statistically significant elevation of caspase 3 activity was observed in spontaneously overactivated eggs (Figure 1D and Figure 2D), further highlighting the difference between overactivation-triggered cell death and apoptosis.

### 2.2. MPF and CSF in Overactivated Eggs

Mature frog eggs are arrested in the metaphase of the second meiotic division due to high activity of MPF and CSF. This can be experimentally confirmed by the presence of M-phase cyclin B2 and the high level of MAPK phosphorylation in these cells (Figure 3). Exit from the meiotic metaphase arrest is triggered by a transient of intracellular calcium after fertilization or parthenogenetic activation, followed by the inactivation of the two meiotic factors. It was demonstrated previously that the oxidative stress-induced overactivation of *Xenopus* eggs is also a calcium-dependent process accompanied by the inactivation of metaphase-specific cyclin [19]. However, it was not verified whether MAPK becomes inactivated in eggs overactivated by oxidative stress. Also, meiotic exit has not been confirmed in spontaneously overactivated frog eggs. The results shown in Figure 3A,B demonstrate that cyclin B is robustly degraded in spontaneously overactivated eggs, as well as in apoptotic eggs. Furthermore, the MAP kinase becomes dephosphorylated in both overactivated and apoptotic eggs (Figure 3A,C). Based on these results, we conclude that *Xenopus* eggs exit the meiotic metaphase arrest upon spontaneous overactivation or apoptosis. Previously, it was reported that apoptosis unfolds in *Xenopus* eggs following exit from the meiotic arrest [6,7]. 

### 2.3. Inducing Egg Overactivation by Mechanical Stress

The spontaneous overactivation of naturally laid frog eggs is a relatively rare phenomenon that typically affects less than 3% of the entire egg population [18]. We have noticed that overactivation occurs most often immediately after egg spawning and suggested that some external factor may promote overactivation in the process of egg deposition. The most plausible candidate for this factor would be mechanical stress, as egg spawning is accompanied by significant constriction and strain in the genital tract. To confirm the involvement of mechanical stress in egg overactivation, ovulated frog eggs were subjected to repeated pipetting through a cut plastic pipette tip (see Section 4 for details). It was found that the pipetting resulted in the appearance of eggs bearing the distinctive morphological features of overactivation, such as white coloring and an increased cell diameter (Figure 4). Importantly, these eggs emerged immediately after pipetting (Figure 4A), indicating that mechanical stress induced egg overactivation. In addition, the effect of pipetting was dose-dependent; larger numbers of pipetting cycles produced more damaged eggs (Figure 4A). Furthermore, ATP depletion, a hallmark of egg overactivation, was also observed in eggs subjected to mechanical stress (Figure 4D). Of note, as with spontaneous overactivation, caspase activity was not elevated in these eggs (Figure 4E). Next, we investigated whether exit from the meiotic metaphase arrest occurred in eggs overactivated by mechanical stress. It was found that meiotic cyclin B was rapidly degraded, but the MAP kinase remained phosphorylated in these eggs (Figure 5). This result partially contradicts the findings obtained with spontaneously overactivated eggs, where both cyclin B degradation and MAP kinase dephosphorylation were observed (Figure 3). A probable explanation for this discrepancy is provided in the Section 3. Taken together, the data presented in Figure 4 and Figure 5 demonstrate that mechanical stress causes the overactivation of spawned *Xenopus* eggs, suggesting that it can be a factor in promoting overactivation during oviposition, as debated further in the Section 3.

### 2.4. Changes in ATP and ADP Contents in Overactivated Eggs

The fast drop in the level of intracellular ATP observed in the overactivated eggs raised the question about the probable mechanism of the ATP depletion. We hypothesized that both intracellular consumption and ATP release from eggs might be involved in this phenomenon. To address this issue, ATP levels, as well as ADP/ATP ratios, were determined inside and outside overactivated and untreated (control) eggs. In accordance with previously presented results, it was found that ATP was largely depleted from both spontaneously overactivated eggs and eggs overactivated by mechanical stress but not from control non-activated eggs (Figure 6A). Importantly, considerable amounts of leaked ATP were detected outside the overactivated eggs but not the control eggs. This result indicates that plasma membrane permeability is greatly increased in overactivated eggs, allowing for the outflow of intracellular ATP. Considering the extremely robust ATP discharge, it is reasonable to suggest that plasma membrane integrity is compromised in these cells. Furthermore, it was found that the ADP/ATP ratio was drastically increased in the overactivated eggs, whereas it remained low in the control cells. The ratio was also highly elevated in the extracellular compartment of the overactivated eggs, albeit to a lesser extent than inside the eggs (Figure 6B). Of note, the amount of ATP released from the eggs overactivated by mechanical stress was significantly higher than in the case of spontaneous overactivation (Figure 6A). In addition, both the intracellular and extracellular ADP/ATP ratios were higher in the spontaneously overactivated eggs than in the eggs overactivated by mechanical stress (Figure 6B). The implications and significance of these findings are considered further in Section 3.

### 2.5. Egg Overactivation in the Frog Genital Tract

A vast majority of ovulated *Xenopus* eggs are normally laid out within 12 to 18 h after hCG injection. However, up to 10% of eggs still remain in the frog’s body for a much longer time [7]. It was reported that eggs completely disappear from the frog’s genital tract within several days following hormonal stimulation. The quality of the retained eggs worsens progressively, and they deteriorate in the genital tract mainly by caspase-dependent apoptosis [7]. However, it was found in the present study that some eggs retained in the genital tract over 24 h after hCG injection exhibited the distinctive morphological features of overactivated eggs, such as white coloring and an increased cell diameter (Figure 7A,B). In addition, ATP depletion, a hallmark of egg overactivation, was also observed in the retained eggs (Figure 7C). Of note, apoptotic eggs retained in the genital tract displayed only a moderate decrease in intracellular ATP (Figure 7C) and significant caspase activation (Figure 7D), distinguishing apoptosis from overactivation-induced cell death. These data indicate that egg overactivation can infrequently take place in the frog’s genital tract alongside apoptosis.

## 3. Discussion

In the present study, we investigated a minor subpopulation of overactivated eggs consistently observed among spawned frog eggs. Several types of cells can clearly be distinguished by their appearance in aging populations of ovulated unfertilized *Xenopus* frog eggs. They include mature fertilization-competent eggs, activated eggs with the contracted pigment layer, decoloring apoptotic eggs, and entirely white overactivated eggs [18]. Notably, in contrast to reversible actin/myosin-based and calcium/protein kinase C-mediated cortical contraction observed in fertilized or parthenogenetically activated *Xenopus* eggs [20], cortical contraction in overactivated eggs is irreversible, resulting in the persistent white color of these eggs (Figure 1, Figure 2, Figure 4 and Figure 7). During egg activation, the internal cortex layer containing pigment granules changes its supramolecular organization and relocates temporarily to the egg’s interior. After a short calcium transient, which ends within about 10 min in fertilized or parthenogenetically activated *Xenopus* eggs, the internal pigmented cortex layer restores its original organization. However, calcium is constitutively elevated in overactivated eggs [17], making cortical contraction in these eggs irreversible and permanent. Presently, it is not known whether the pigment granules deteriorate or whether the pigment becomes degraded at the late stages of overactivation. Further ultrastructural and biochemical studies are necessary to reveal the distribution of the pigment granules and other oocyte organelles in overactivated eggs.

The proportion of overactivated eggs is quite low in natural egg populations, normally not exceeding 2–3% [18]. Although the natural triggers of egg overactivation are unknown, it was demonstrated previously that strong oxidative stress is capable of inducing an overactivated phenotype. The hallmark biochemical events of oxidative stress-induced overactivation have been recently characterized [18,19]. Now, one of the main findings of our present study asserts that the major events of spontaneous overactivation, such as irreversible cortical contraction, an increase in egg size, the depletion of intracellular ATP, an increase in the intracellular ADP/ATP ratio, and the degradation of M phase-specific cyclin B2, are identical to the previously described events of the oxidative stress-induced process. In addition, as with oxidative stress-induced overactivation, the events of spontaneous overactivation manifest in eggs within just one hour in the absence of caspase activation. Furthermore, the same events unfold in eggs overactivated by mechanical stress (Figure 4, Figure 5 and Figure 6). These findings suggest that an identical physiological scenario develops in eggs overactivated by various means. 

Then, what kind of scenario is this? The finding that ATP is acutely depleted in overactivated eggs strongly suggests necrosis. It was previously proposed that intracellular levels of ATP determine a distinctive way of cell death by apoptosis or necrosis [21] and further demonstrated that ATP depletion can alter the mode of cell death [22,23]. Several studies revealed that necrosis develops in cells treated with drugs or subjected to electric shock under conditions of intracellular ATP depletion [24,25,26]. The phenomenon of ATP depletion was universally observed in eggs overactivated by oxidative stress [18] or mechanical stress (Figure 4 and Figure 5), as well as in eggs overactivated spontaneously outside the frog (Figure 1 and Figure 2) or in the genital tract (Figure 7). The fast depletion of intracellular ATP is one hallmark of classical necrosis that distinguishes it from other types of cell death [27]. By comparison, a decrease in intracellular ATP occurs quite late in apoptosis because high ATP levels are necessary to maintain this process [27,28]. In particular, apoptosome assembly, which is responsible for caspase activation, requires cytochrome C and ATP/dATP binding. However, the events of egg overactivation occur in the absence of caspase activation (Figure 1, Figure 2, Figure 4 and Figure 7), and they unfold much faster than the classical apoptotic process described previously in *Xenopus* eggs [6,7]. Similarly to apoptosis, autophagy, the other major mechanism of programmed cell death activated in response to cellular damage, also requires ATP at different steps of the process [29].

Even more straightforward evidence for a necrotic process in overactivated frog eggs comes from the finding that ATP and ADP are rapidly released from these cells (Figure 6). This finding indicates that plasma membrane integrity is compromised in overactivated eggs. It was reported that ATP can be released from cells in response to various cell-death inducing conditions such as hypoxia, cytotoxic agents, hypertonic shock, plasma membrane damage, etc. [26,30,31,32,33]. Membrane damage was found to trigger robust ATP release in sea urchin embryos [34]. Of note, in our study, overactivated frog eggs were found to lose their ATP not only due to leakage but also due to the intracellular conversion of ATP to ADP, as observed by a greatly increased ADP/ATP ratio in these cells (Figure 6). Previously, the fast metabolic depletion of ATP has been observed in different types of somatic cells dying by necrosis [35]. It was reported that necrosis fosters mitochondrial dysfunction due to inner membrane depolarization, mitochondrial swelling, and rupture, resulting in the loss of ATP production [36]. Mitochondrial damage that occurs in oocytes during their long life span is recognized as one of the most important factors that adversely affect fertility in humans [37].

Furthermore, a significant increase in the diameter of overactivated eggs (Figure 1, Figure 2, Figure 4 and Figure 7) indicates that membrane permeability is augmented and cellular osmotic homeostasis is lost in these eggs. It was reported previously that necrotic cell death is characterized by cell swelling, termed as necrotic volume increase (NVI), in several somatic cell lines [35]. For instance, the chemical anoxia of glial cells leads to surface blebbing and cell swelling to 200% of control values within 20 min [38]. Also, hepatocyte necrosis caused by anoxia or oxidative stress was accompanied by Na+ overload and cell swelling that developed in two phases. It has been suggested that the influx of extracellular Na+ that occurs upon cell injury is a major determinant of NVI [39]. Thus, the cell size increase observed in the present study provides additional evidence for a necrotic process in overactivated frog eggs.

Importantly, although plasma membrane rupture conventionally identifies necrotic cell death, it is debatable whether egg overactivation represents the classical scenario of accidental cell death defined recently by the Nomenclature Committee on Cell Death as “virtually instantaneous and uncontrollable form of cell death corresponding to the physical disassembly of the plasma membrane caused by extreme physical, chemical or mechanical cues” [40]. Firstly, while extremely fast and robust, egg overactivation is not exactly instantaneous since morphological and biochemical changes induced by overactivation develop gradually over time. As a consequence, physical damage of the plasma membrane does not lead to the immediate and massive release of intracellular content, and overactivated eggs maintain their form and size for some time. Secondly, it seems that infringement of the plasma membrane can be caused not only by extreme cues but also by micro-perturbations to the intracellular or extracellular environment, as it evidently takes place in the case of spontaneous overactivation. Thirdly, egg overactivation is not an entirely uncontrollable event, as it can be modulated to some extent by calcium chelators, as demonstrated previously [19]. Therefore, egg overactivation develops progressively in a sequential manner, suggesting that it can be a form of physiologically regulated and ordered cell death.

Our data demonstrate that mechanical stress can cause the overactivation of *Xenopus* eggs (Figure 4, Figure 5 and Figure 6), and we further suggest that it may be a key factor promoting overactivation during egg deposition. The hypothesis that mechanical stimulation can trigger egg overactivation was informed by previous observations. Firstly, evidence that physical deformation during egg oviposition in Hymenoptera can initiate egg activation and embryo development was presented. Eggs of the wasp *Pimpla turionellae* squeeze through a narrow capillary to become activated and develop into male larvae [41,42]. Secondly, it was found that the application of hydrostatic pressure or the manual pulling of dorsal appendages initiated the resumption of meiosis in *Drosophila* oocytes [43,44]. It was further demonstrated that mechanical stimulation during ovulation triggered *Drosophila* egg activation via an influx of calcium into the eggs [45]. Thirdly, it was shown that unfertilized eggs of the frog *Eleutherodactylus coqui* can be easily activated by mechanical stimulation. It was proposed that spontaneous activation, observed in 34% of eggs, occurs in response to mechanical stress during oviposition [46]. Finally, *Xenopus* eggs can be activated by mechanical stimulation, such as pricking with a sharp needle. A wave of free cytosolic calcium is initiated starting from the point of prick activation [47,48]. Of note, overactivation and activation were not distinguished in the abovementioned studies. Thus, based on previous reports and the results obtained in the present study, we suggest that mechanical stress during oviposition may promote the overactivation of *Xenopus* eggs. 

Apparently, there still exist some differences between spontaneous and stress-induced overactivation. For instance, the amount of leaked ATP is greater, and the ADP/ATP ratio of both extracellular and intracellular content is lower in eggs overactivated by mechanical stress as compared to spontaneously overactivated eggs (Figure 6). These facts can be explained by greater damage to the plasma membrane caused by mechanical stress, leading to a faster leakage of ATP before its conversion to ADP by intracellular enzymes. Furthermore, although cyclin B is degraded both in eggs overactivated by mechanical stress and spontaneously overactivated eggs, MAPK becomes dephosphorylated only in the latter case (Figure 3 and Figure 5). This discrepancy can also be explained by the different degrees of plasma membrane damage in cells overactivated by different means. Cyclin degradation is initiated by a calcium transient upon egg overactivation [19], and it occurs before the downregulation of the MAPK pathway in activated *Xenopus* eggs [49,50]. A relatively mild breach of the plasma membrane in spontaneously overactivated eggs keeps the positive feedback between MPF and CSF [51] operational long enough to bring about CSF inactivation and MAPK dephosphorylation. However, more severe membrane damage in stress-overactivated eggs disrupts this feedback rapidly, making MAPK dephosphorylation impossible. Thus, different severities of plasma membrane damage can account for the observed minor differences between spontaneous and stress-induced overactivation. 

Markedly, although spontaneous overactivation is a relatively rare phenomenon, normally affecting only a small minority of ovulated eggs, our data demonstrate that mechanical or oxidative stress can greatly increase the frequency of overactivation. In fact, egg overactivation becomes practically inevitable under high oxidative stress [18]. Thus, it can be hypothesized that, in some cases, the proportion of overactivated eggs in natural egg populations may increase significantly, reflecting egg condition and stress intensity. Considering that mechanical stress accompanies oviposition in different species (see Section 3 above), the increased manifestation of egg overactivation under stress-inducing conditions requires additional investigation. Approaches that can prevent or attenuate overactivation should be pursued with the aim of increasing egg quality. At present, it is not known whether mammalian eggs can experience overactivation; previous studies have not discriminated between egg activation and overactivation. However, if that is the case, the findings in frogs can possibly be extended to mammalian eggs with applications in assisted reproduction. In addition, studies of overactivated eggs could expand our understanding of cell death by disclosing alternative physiological mechanisms. Further investigations are necessary to delineate in detail intracellular molecular events in overactivated eggs.

## 4. Materials and Methods

### 4.1. Reagents

Water-soluble progesterone (PG), anesthetic MS-222, and an ATP Bioluminescence Assay Kit CLS II were purchased from Sigma (St. Louis, MO, USA). Human chorionic gonadotropin (hCG) was obtained from Teikoku Zoki (Tokyo, Japan), and collagenase (280 U/mg) was obtained from Wako (Osaka, Japan). Fluorogenic caspase-3 substrate IV was purchased from Calbiochem (La Jolla, CA, USA), and Apo-ONE homogenous caspase 3/7 assay was obtained from Promega (Madison, WI, USA). Polyclonal anti-cyclin B2 antibody was ordered from Santa Cruz (Dallas, TX, USA), and biotinylated anti-rabbit IgG was purchased from Vector Laboratories (Burlingame, CA, USA). Polyclonal anti-MAPK and anti-pMAPK antibodies were purchased from Cell Signaling (Beverly, MA, USA). The Streptavidin Biotin Complex Peroxidase Kit and CBB protein assay kit were obtained from Nacalai Tesque (Kyoto, Japan), and the bioluminescent ApoSENSOR ADP/ATP ratio assay kit was purchased from BioVision (Mountain View, CA, USA). Other chemicals were obtained from Wako and Nacalai Tesque.

### 4.2. Animals and Cells

Adult wild-type female *Xenopus laevis* frogs were purchased from Shimizu (Kyoto, Japan) and maintained in dechlorinated water at an ambient temperature of 21–23 °C. The experiments with the animals were conducted according to the Kyoto Sangyo University Animal Experimentation Regulations under the permission N 2018-20. The experiments with oocytes and eggs were carried out at an ambient temperature of 21–23 °C.

Egg ovulation was induced by the injection of 500 U/animal of human chorionic gonadotropin in the dorsal lymph sacs of the female frogs. Eggs were collected by gently squeezing the abdomen at about 12 h after injection and kept in OR-2 buffer at ambient temperature. It is acknowledged that eggs obtained by the hormone treatment and abdominal pressure are identical to naturally laid eggs. Immature oocytes were isolated as detailed previously [6]. Briefly, the frogs were anesthetized in 2 mg/mL solution of MS-222; then, the ovaries were surgically removed and placed into OR-2 solution containing 82.5 mM NaCl, 2.5 mM KCl, 1 mM CaCl_2_, 1 mM MgCl_2_, 1 mM Na_2_HPO_4_, and 5 mM HEPES, pH 7.6. The ovaries were manually dissected into clumps of 50–100 oocytes and extensively washed with OR-2 solution. Oocytes were treated with 5 mg/mL collagenase in OR-2 at 21 °C for 3 h by shaking at 60 rpm, extensively washed in OR-2 solution, and left for stabilization over 4 h. Undamaged defolliculated stage VI oocytes ranging in size from 1.2 to 1.3 mm were manually selected and used in the experiments. In vitro oocyte maturation was induced by the addition of 5 μM PG and monitored by the appearance of a white spot on the animal hemispheres of the oocytes. To obtain crude cytosolic fractions, the eggs were homogenized by pipetting in a ten-fold volume of cold OR-2 buffer containing protease inhibitors APMSF and leupeptin and then centrifuged at 10,000 rpm, 4 °C, for 10 min. Supernatant fractions were collected and stored on ice until biochemical analysis.

### 4.3. Microscopic Observations

Observation and imaging of the *Xenopus* eggs were carried out using an SZX16 stereo zoom microscope (Olympus, Tokyo, Japan) equipped with a high-frame digital microscope CCD camera DP73, CCD interface U-TV0.5XC-3, wide-angle objective SDF PLAPO 1xPF. The CellSens Standard software (product version): standard 1.8.1 (Olympus, https://www.olympus-lifescience.com/en/software/cellsens/, accessed on 1 May 2024) was used for image acquisition. The acquired images were further processed with the ImageJ ver. 1.53a software of the National Institute of Health (https://imagej.net/ij/, accessed on 1 May 2024) [52] freely available at https://imagej.nih.gov/ij/ (accessed on 1 May 2024).

### 4.4. Mechanical Stress

Mature ovulated eggs surrounded by a jelly layer were subjected to repeated pipetting through a 1 mL plastic pipette tip. The tip for pipetting was cut off at the end and tempered softly with fire to attain the size of the tip opening that mildly strained the eggs in the process of pipetting. The treated eggs were kept in OR-2 solution and monitored for the appearance of morphological features of overactivation over 2 h.

### 4.5. Measurement of Intracellular and Extracellular ATP and ATP/ADP Ratio

Measurements of intracellular and extracellular ATP and the ADP/ATP ratio were carried out using the ATP Bioluminescence Assay Kit CLS II and ApoSENSOR assay kit according to the manufacturer’s manuals. Eggs were incubated in a one-hundred-fold volume of OR-2 solution. The crude cytosolic fractions were obtained by egg homogenization in a one-hundred-fold volume of 50 mM Tris buffer, pH 7.5, containing 4 mM EDTA, followed by centrifugation at 10,000 rpm, 4 °C, for 10 min. Then, 1 µL aliquots of supernatant or incubation solution were used in 100 µL bioluminescence assays. Luminescence intensity was quantified using the GeneLight GL-220 portable luminometer (Microtec, Funabashi, Japan) within one minute after the initiation of the luciferase reaction by sample addition.

### 4.6. Immunoblotting

To monitor cyclin B2 content and MAPK phosphorylation status, crude cytosolic fractions of oocytes and eggs were heated at 95 °C for 5 min in the presence of SDS–sample buffer (62.5 mM Tris–HCl, pH 6.8, 2% SDS, 10% sucrose, 0.01% BPB, 100 mM DTT). Protein samples were separated by SDS PAGE using 10% polyacrylamide gels and transferred to PVDF membranes using a semidry blotting device from BioRad (Hercules, CA, USA). The membranes were blocked with T–TBS buffer (20 mM Tris–HCl, pH 7.5, 150 mM NaCl, 0.05% Tween 20) containing 3 mg/mL bovine serum albumin and incubated at room temperature for 2 h with the 200-fold diluted anti-cyclin B2 antibody, or 100-fold diluted anti-phospho MAPK, or 200-fold diluted anti-MAPK antibodies. The membranes were extensively washed with T-TBS buffer and treated with 1000-fold-diluted biotinylated anti-rabbit IgG, then with peroxidase-conjugated streptavidin, as per the manufacturer’s manual for the Streptavidin Biotin Complex Peroxidase Kit. The immune complexes were detected by color development catalyzed by peroxidase in the presence of hydrogen peroxide and diaminobenzidine tetrahydrochloride.

### 4.7. Other Methods

Protein content in the egg cytosolic fractions was determined with the CBB protein assay. Sample absorbance was measured using a NanoDrop 1000 Spectrophotometer (Thermo Fisher Scientific, Waltham, MA, USA). Bovine serum albumin was employed as a calibration standard. Caspase activity assay was performed using caspase-3 or caspase 3/7 substrates as described previously [6]. The quantified data in figures are presented as the means ± SD values of four to six replicates. The experiments were repeated with separate batches of eggs obtained from at least three different animals.

## 5. Conclusions

Egg overactivation is an abnormal physiological process that occasionally accompanies ovulation and spawning in frogs. In the present study, the major biochemical and cytological events of spontaneous overactivation were investigated in unfertilized metaphase II-arrested *Xenopus* frog eggs. Our study demonstrates that (i) irreversible cortical contraction, an increase in egg size, the depletion of intracellular ATP, a sharp increase in the intracellular ADP/ATP ratio, the degradation of M phase-specific cyclin B2, and MAPK dephosphorylation occur in the eggs within one hour of spontaneous overactivation; (ii) the observed events develop in the absence of caspase activation; (iii) mechanical stress can cause the overactivation of meiotically arrested frog eggs; (iv) egg overactivation can occur in the frog’s genital tract; and (v) plasma membrane integrity is compromised in overactivated eggs. Our study demonstrates that the major events of spontaneous and mechanical stress-induced overactivation are largely identical to the previously described events of the oxidative stress-induced process. The observed biochemical and morphological changes indicate that overactivated eggs die by necrosis.

## Figures and Tables

**Figure 1 ijms-25-05321-f001:**
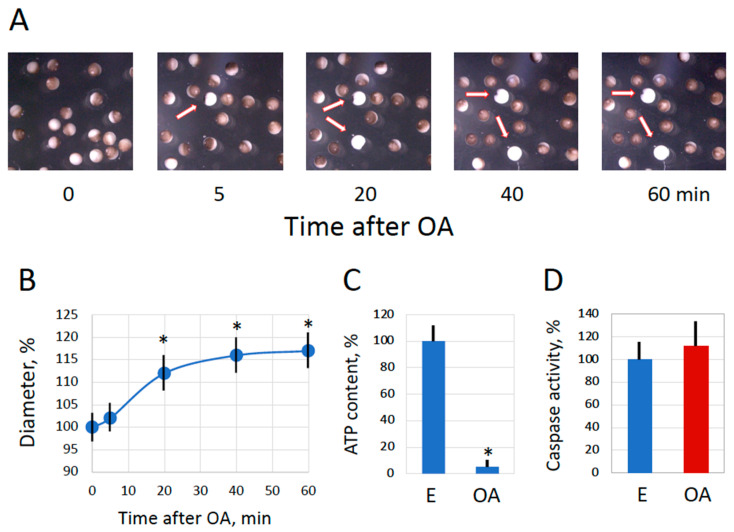
Spontaneous overactivation of ovulated *Xenopus* eggs. Phenotypical changes in a subpopulation of spawned *Xenopus* eggs are shown in panel (**A**). Time “0” refers to the time of egg deposition; the arrows in the panels point to spontaneously overactivated eggs. The progressive increase in the diameter of overactivated eggs and evaluation of intracellular ATP in normal (E) and overactivated (OA) eggs are presented in panels (**B**,**C**), respectively. The quantification of caspase activity is shown in panel (**D**). The OA eggs in panels (**C**,**D**) were analyzed one hour after the visually identified beginning of overactivation. The bars in panels (**B**–**D**) represent standard deviation, and the asterisks in panels (**B**,**C**) indicate statistical difference from the control eggs (*p* < 0.05). Around one thousand eggs from three independent batches were monitored in panel (**A**), ten normal and ten overactivated eggs were measured at each time point in panel (**B**), and five eggs of each cell type were analyzed in panels (**C**,**D**).

**Figure 2 ijms-25-05321-f002:**
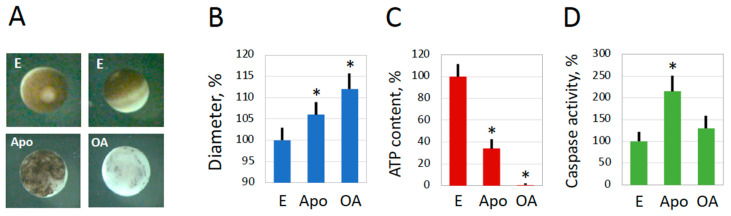
Spontaneous overactivation of in vitro-matured frog eggs. Morphological types of metaphase-arrested (E), overactivated (OA), and apoptotic (Apo) eggs observed in a subpopulation of defolliculated in vitro-matured *Xenopus* eggs at 24–30 h after addition of 5 μM PG are highlighted in panel (**A**). Measurements of egg diameter and intracellular ATP are presented in panels (**B**,**C**), respectively. Quantification of caspase activity in analyzed cell types is presented in panel (**D**). Mature overactivated eggs were analyzed one hour after the visually identified beginning of overactivation; metaphase-arrested and apoptotic eggs were examined at 24–30 h after hormone administration. The bars in panels (**B**–**D**) represent standard deviation, and the asterisks indicate statistical differences from the control eggs (*p* < 0.05). Around one thousand eggs from three independent batches were monitored in panel (**A**), ten eggs of each cell type were measured in panel (**B**), and five eggs of each cell type were analyzed in panels (**C**,**D**).

**Figure 3 ijms-25-05321-f003:**
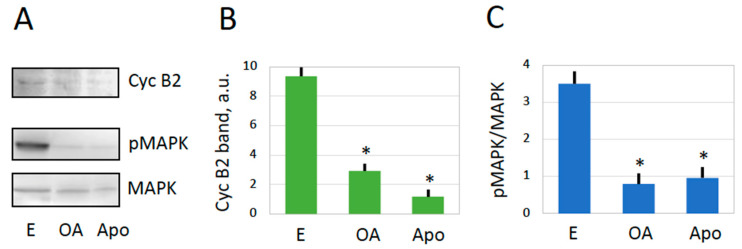
Meiotic exit after spontaneous overactivation. An evaluation of cyclin B2 contents and phosphorylation state of the MAP kinase in metaphase-arrested (E), overactivated (OA), and apoptotic (Apo) eggs is presented. Panel (**A**) shows representative immunoblots, and data quantification is presented in panels (**B**,**C**). Eggs were matured in vitro in the presence of 5 μM progesterone and examined 24–30 h after hormone administration. The immunoblotting samples contained a normalized amount of total protein, 50 μg per lane. The experiment was repeated with three separate batches of eggs, and the results of a single-bath experiment are shown. The bars in panels (**B**,**C**) represent standard deviations obtained in four measurements, and the asterisks indicate statistical differences from the control eggs (*p* < 0.01). More than five cells of each cell type were subjected to the analysis.

**Figure 4 ijms-25-05321-f004:**
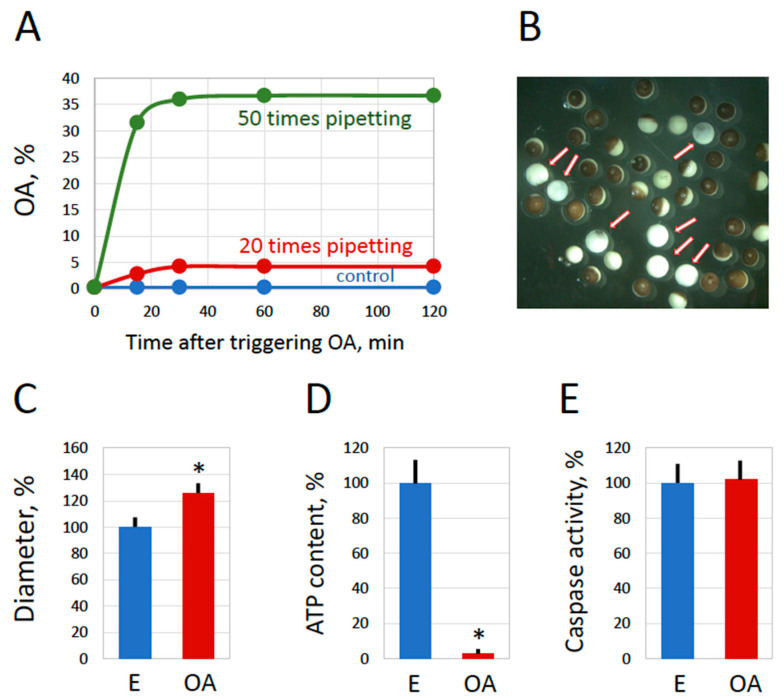
Mechanical stress induces egg overactivation. Overnight-ovulated frog eggs surrounded by a jelly layer were subjected to repeated pipetting through a cut plastic pipette tip. The emergence of overactivated eggs within 2 h after pipetting 20 (red line) or 50 (green line) times is presented in panel (**A**). A mixed population of normal and overactivated eggs observed 2 h after pipetting is shown in panel (**B**). The arrows in the panel point to overactivated eggs. Measurements of the egg diameter and intracellular ATP content performed in 2 h after pipetting are presented in panels (**C**,**D**), respectively. Panel (**E**) shows the quantification of caspase activity in control and overactivated eggs. The bars in panels (**C**–**E**) represent standard deviations. The asterisks in panels (**C**,**D**) indicate statistical differences from the control eggs (*p* < 0.05). More than five cells of each morphological type were analyzed in panels (**C**–**E**).

**Figure 5 ijms-25-05321-f005:**
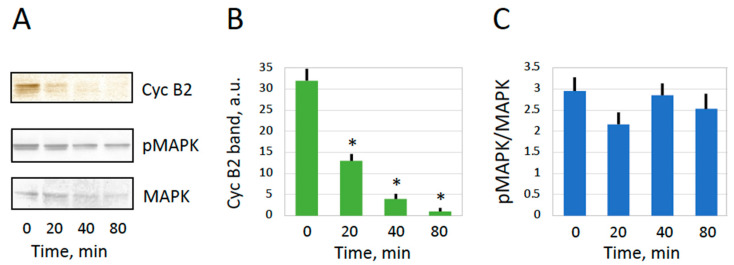
Cyclin B2 content and MAPK phosphorylation in eggs overactivated by mechanical stress. Cyclin B2 content and the phosphorylation state of the MAP kinase were analyzed at different times after the visually identified beginning of overactivation. Panel (**A**) shows representative immunoblots, and data quantification is presented in panels (**B**,**C**). The immunoblotting samples contained a normalized amount of total protein, 50 μg per lane. The eggs were matured in vitro in the presence of 5 μM progesterone and utilized 12 h after hormone administration. The experiment was repeated with three separate batches of eggs obtained from different animals, and the results of a single-bath experiment are shown. The bars in panels (**B**,**C**) represent standard deviations obtained in four measurements, and the asterisks indicate statistical differences from the control eggs (*p* < 0.01). More than five cells were analyzed at each time point.

**Figure 6 ijms-25-05321-f006:**
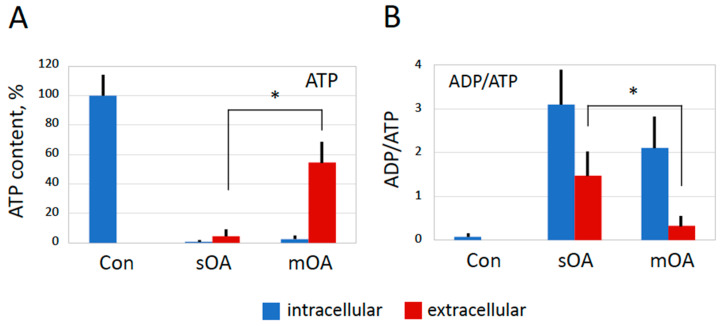
Changes in ATP and ADP levels elicited by egg overactivation. The relative contents of ATP inside and outside spontaneously (sOA) and mechanically (mOA) overactivated eggs, as well as untreated control eggs (Con), are presented in panel (**A**). Intracellular and extracellular ADP/ATP ratios are evaluated in panel (**B**). Data for intracellular and extracellular compartments are presented in blue and red, respectively. All measurements were performed one hour after the visually identified beginning of overactivation. The experiment was repeated with four different eggs of each cell type (Con, sOA, and mOA). The bars in the panels represent standard deviations, and the asterisks indicate statistical significance at *p* < 0.01.

**Figure 7 ijms-25-05321-f007:**
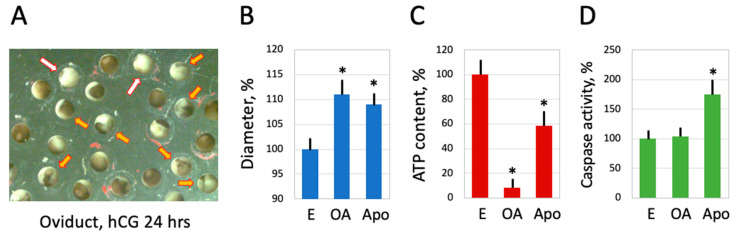
Occurrence of egg overactivation in the frog genital tract. A subpopulation of *Xenopus* eggs retained in the genital tract for 30 h after hCG injection is shown in panel (**A**). The white and orange arrows in the panel point to overactivated (OA) and apoptotic (Apo) eggs, respectively. Measurements of egg diameter and intracellular ATP content are presented in panels (**B**,**C**), respectively. The quantification of caspase activity in the analyzed cell types is presented in panel (**D**). The bars in panels (**B**–**D**) represent standard deviations. The asterisks in panels (**B**–**D**) indicate statistical differences from the control eggs (*p* < 0.05). More than five cells of each morphological type were analyzed in the panels.

## Data Availability

Data are contained within the article.

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
