# Peer review of "Spontaneous Overactivation of *Xenopus* Frog Eggs Triggers Necrotic Cell Death"

_ijms, 2024, doi:10.3390/ijms25105321_

Round 1

Reviewer 1 Report (Previous Reviewer 2)

Comments and Suggestions for Authors

This is important for the egg fertilization field paper. It is well-written and well-documented.

 However, several points need to be addressed.

  1. Authors state several times that the overactivated eggs increase in size but do not explain why and do not discuss the size increase subject.  
  2. What is the explanation for the size increase? 
  3. Are there any other examples of size increase (and mechanisms) for the dying cells? 
  4. Also, the manuscript would benefit from including a higher magnification of the control and overactivated eggs.
  5. Another issue that is not covered in the manuscript is the white color of overactivated eggs. What happens to the pigment? Does it relocate from the cortex to the egg interior? Or the pigment becomes degraded?
  6. It would be good to fix and section the oocytes and compare the distribution of pigment, yolk, and other oocyte organelles, and include these section images in the paper
Comments on the Quality of English Language

minor grammar issues

Author Response

Response to reviewers’ reports

We would like to thank the reviewers for their constructive criticism and valuable comments that helped us to improve the manuscript. The detailed point-by-point response to the points raised is following.

Reviewer 1

The English language was corrected throughout the entire manuscript by native speaking professionals.

1. Authors state several times that the overactivated eggs increase in size but do not explain why and do not discuss the size increase subject.

2. What is the explanation for the size increase?

3. Are there any other examples of size increase (and mechanisms) for the dying cells?

The size increase during cell death is an important subject, that has been previously addressed by several studies. This subject is much relevant to the present study. To address this issue, we have added a new paragraph in Discussion (lines 402-412), providing the examples of cell size increase in dying cells and suggested mechanism of this phenomenon, as follows:

“Furthermore, a significant increase in the diameter of overactivated eggs (Figures 1, 2, 4, 7) indicates that membrane permeability is augmented and cellular osmotic homeostasis is lost in these eggs. It was reported previously that necrotic cell death is characterized by cell swelling, termed as necrotic volume increase (NVI), in several somatic cell lines [36]. For instance, chemical anoxia of glial cells leads to surface blebbing and cell swelling to 200% of control values within 20 minutes [37]. Also, hepatocyte necrosis caused by anoxia or oxidative stress was accompanied by Na+ overload and cell swelling that developed in two phases. It was suggested that the influx of extracellular Na+ that occurs upon cell injury is a major determinant of NVI [38]. Thus, the cell size increase observed in the present study provides additional evidence for a necrotic process in overactivated frog eggs.”

4. Also, the manuscript would benefit from including a higher magnification of the control and overactivated eggs.

Thanks for the valuable suggestion. As recommended, we have introduced the higher magnification images in Figure 2A.

5. Another issue that is not covered by in the manuscript is the white color of overactivated eggs. What happens to the pigment? Does it relocate from the cortex to the egg interior? Or the pigment becomes degraded?

To address this issue, we have added a new paragraph in Discussion (liness 344-357), as follows:

“Notably, in contrast to reversible actin/myosin-based and calcium/protein kinase C-mediated cortical contraction observed in fertilized or parthenogenetically activated Xenopus eggs [21], cortical contraction in overactivated eggs is irreversible, resulting in the persistent white color of these eggs (Figures 1, 2, 4, 7). During egg activation, the internal cortex layer containing pigment granules changes its supramolecular organization and relocates temporarily to the eggs interior. After a short calcium transient, which ends within about 10 minutes in fertilized or parthenogenetically activated Xenopus eggs, the internal pigmented cortex layer restores its original organization. However, calcium is constitutively elevated in the overactivated eggs [17], making cortical contraction in these eggs irreversible and permanent. Presently, it is not known whether the pigment granules deteriorate and the pigment becomes degraded at the late stages of overactivation. Further ultrastructural and biochemical studies are necessary to reveal distribution of the pigment granules and other oocyte organelles in overactivated eggs.”

6. It would be good to fix and section the oocytes and compare the distribution of pigment, yolk and other oocyte organelles, and include these section images in the paper.

We appreciate this suggestion. Indeed, egg sectioning can provide additional information not only on the distinctive pigmentation, but also on the fate of other egg organelles in overactivated frog eggs. However, the main purpose of our present work is to provide evidence for necrosis of overactivated eggs, and the ultrastructural studies, although very related and important, can add little to this issue. Still, we think that this subject warrants additional extensive studies in the future. This perception is reflected in the ending part of the above-mentioned paragraph (lines 353-357), as follows:

“Presently, it is not known whether the pigment granules deteriorate, and the pigment becomes degraded at the late stages of overactivation. Further ultrastructural and biochemical studies are necessary to reveal distribution of the pigment granules and other oocyte organelles in overactivated eggs.”

In addition, the English language was corrected throughout the entire manuscript by native speaking professionals.

Reviewer 2 Report (Previous Reviewer 1)

Comments and Suggestions for Authors

Spontaneous overactivation of Xenopus frog eggs triggers necrotic cell death

This manuscript is a resubmitted, adapted version of a manuscript that I had previously reviewed. Although the authors have answered some of my questions satisfactorily and changed the manuscript accordingly, there are still certain aspects that make this manuscript not suitable for publication in IJMS.

In the materials and methods: ‘Egg ovulation was induced by injection of 500 U/animal of human chorionic gonadotropin in the dorsal lymph sac of female frogs. Eggs were collected by squeezing abdominal parts of the animals’.  This does not appear to be ‘spontaneous’. Related to that, the eggs shown in Figure 1 are according to the figure legend ‘naturally laid’. Does that mean without hormone treatment and squeezing?

Statistics: Although in graphs standard deviations have been added, the graphs of figures 3, 5 and 6 do not show which bars significantly are different from one another (for instance with asterisks as has been done in figures 1,2, 4 and 7.

Statistics: In the figure legend a standard deviation of the mean is mentioned (SD values of means), which seems odd. It is either standard deviation, or standard error of the mean.

Figure 5A,B. Quantified levels of CycB2 expression. Although in contrast to the previous version, the levels of Cyc B2 are now normalized to total protein content, apposed to levels of a non-specific protein, the values are exactly the same. How can this be?

Line 321: ‘a lot of eggs still remain in the frog’s body for much longer’ This is rather nonspecific. What is ‘a lot’ and what is ‘much longer. This should be more specific.

Although the authors give some examples, the physiological relevance, and the comparison with mammalian eggs seem far-fetched. Examples are provided of insect eggs, which have a different physiology and development compared with amphibian eggs. Also, ovulation in mammals cannot be compared with egg deposition in Xenopus laevis.

At the end of the discussion at line 450 it reads ‘ egg overactivation becomes practically inevitable under high stress (Figure 4)’. However, in figure 4 it is demonstrated that even after 20x pipetting still only 5% of the eggs demonstrate overactivation. Even with 50x pipetting, which seems  very stressful indeed, 35% of the cells overactivate, which is still a minority (as 65% of the cells would behave normally). ‘Practically inevitable’ is therefore not supported by the data.

Author Response

Response to reviewers’ reports

We would like to thank the reviewers for their constructive criticism and valuable comments that helped us to improve the manuscript. The detailed point-by-point response to the points raised is following.

Reviewer 2

In the materials and methods: ‘Egg ovulation was induced by injection of 500 U/animal of human chorionic gonadotropin in the dorsal lymph sac of female frogs. Eggs were collected by squeezing abdominal parts of the animals’. This does not appear to be ‘spontaneous’. Related to that, the eggs shown in Figure 1 are according to the figure legend ‘naturally laid’. Does that mean without hormone treatment and squeezing?

It is generally recognized that Xenopus eggs obtained by hormonal stimulation of female frogs and abdominal squeezing are identical to naturally laid eggs. To highlight this point, we have included the following sentence in Materials and Methods (lines 103, 104):

“It is acknowledged that the eggs obtained by hormone treatment and abdominal pressure are identical to naturally laid eggs.”

In addition, we have changed the term “naturally laid eggs” in the figure 1 legend (line 201), in the title of Section 3.1 (lane 165) and in the lines 173, 177 to “spawned eggs”.

Statistics: Although in graphs standard deviations have been added, the graphs of figures 3, 5 and 6 do not show which bars significantly are different from one another (for instance with asterisks as has been done in figures 1, 2, 4 and 7).

We have indicated statistical significance in the graphs of figures 3, 5 and 6 with asterisks.

Statistics: In the figure legend a standard deviation of the mean is mentioned (SD values of means), which seems odd. It is either standard deviation, or standard error of the mean.

Indeed, standard deviation represents, by definition, the deviation of the mean. As suggested, we have changed the expression “SD values of means” to “standard deviation” in all figure legends.

Figure 5A,B. Quantified levels of CycB2 expression. Although in contrast to the previous version, the levels of CycB2 are now normalized to total protein content, apposed to levels of a non-specific protein, the values are exactly the same. How can this be?

The values of quantified levels in figure 5, as well as in figure 3, are arbitrary units (CycB2 band, a.u.). They can be just any values, but most importantly, they should reflect the relative band abundance. Therefore, we have not changed the values, although the normalization method has been changed. 

Line 321: “a lot of eggs still remain in the frog’s body for much longer “. This is rather nonspecific. What is “a lot” and what is “much longer”. This should be more specific.

We have specified the volume of retained eggs in the following sentence (lines 317, 318):

“However, up to 10% of eggs still remain in the frog’s body for much longer time [7].”

Also, the following sentence provides the time span of egg retainment (lines 318, 319):

“It was reported that eggs completely disappear from the frog’s genital tract within several days following hormonal stimulation”.

Although the authors give some examples, the physiological relevance, and the comparison with mammalian eggs seem far-fetched. Examples are provided of insect eggs, which have a different physiology and development compared with amphibian eggs. Also, ovulation in mammals cannot be compared with egg deposition in Xenopus laevis.

The comparison of frog and mammalian eggs may look far-fetched, considering their different physiology and development, however, it is well established that calcium-mediated egg activation occurs universally in all sexually reproducing organisms.

Similarly, the examples of insect eggs are provided here in order to demonstrate that mechanical stress can universally trigger egg activation in different species. Notably, we have also provided the examples of frog eggs in lines 440-445, as follows:

“Thirdly, it was shown that unfertilized eggs of the frog Eleutherodactylus coqui can be easily activated by mechanical stimulation. It was proposed that spontaneous activation, which was observed in 34% of eggs, occurs in response to mechanical stress during oviposition [45]. Finally, Xenopus eggs can be activated by mechanical stimulation, such as pricking with a sharp needle. A wave of free cytosolic calcium is initiated starting from the point of prick activation [46, 47].”   

In addition, to avoid the far-fetched comparison of frog and mammalian eggs, we have rewritten lines 482-486 of Discussion, as follows:

“At present, it is not known whether mammalian eggs can experience overactivation; previous studies have not discriminated between egg activation and overactivation. However, if that is the case, the findings in frogs can possibly be extended to mammalian eggs with applications in assisted reproduction.”

At the end of discussion at line 450 it reads “egg overactivation becomes practically inevitable under high stress (Figure 4)”. However, in figure 4 it is demonstrated that even after 20x pipetting still only 5% of the eggs demonstrate overactivation. Even with 50x pipetting, which seems very stressful indeed, 35% of the cells overactivate, which is still a minority (as 65% of the cells would behave normally). “Practically inevitable” is therefore not supported by the data.

That is true, the data presented in figure 4 do not support “practically inevitable”. However, the full quoted sentence is: “In fact, egg overactivation becomes practically inevitable under high stress (Figure 4), [18].”  Here, the reference [18] concerns oxidative stress-induced overactivation that can reach ~100% at high concentrations of H2O2.

To avoid ambiguity, we have modified the above sentence, as follows (lines 471,472):

“In fact, egg overactivation becomes practically inevitable under high oxidative stress [18].”

Reviewer 2

Comments and Suggestions for Authors

There are some grammatical and other errors for example: " Water-soluble Adult wild-type female frogs Xenopus laevis were purchased from 99 Shimizu (Kyoto, Japan) and maintained in dechlorinated water at the ambient 100 temperature of 21–23°C." The whole text should be checked.

The reviewer is right.

“Water-soluble Adult wild-type female frogs Xenopus laevis” sounds at least exotic! It is very difficult to imagine water-soluble frogs. What a mistake!

The sentence has been rewritten. (lines 98, 99)

In addition, the whole manuscript text has been checked thoroughly.

It is very unclear to me why the authors studied the spontaneous egg over activation if as they admitted: "Normally, the proportion of overactivated eggs is quite low in natural egg  populations; it rarely exceeds several percent" .  What is the overall meaning of their findings? Also when you damage the eggs by vigorous pipetting no wander that these eggs die, I would not call it a mechanical stress comparable to the possible mechanical stress in the genital tract.

Physiological relevance of egg overactivation and the possible role of mechanical stress in this process are very important issues of this study. They were also raised by reviewer 1. To address these issues, we have introduced two new paragraphs in Discussion (lines 408-427 and 447-462), as follows:

“Our data demonstrate that mechanical stress can cause overactivation of Xenopus eggs (Figures 4-6), and we further suggest that it may be a key factor that promotes overactivation during egg deposition. The hypothesis that mechanical stimulation can trigger egg overactivation was suggested by previous observations. First, evidence was presented that, physical deformation during egg oviposition in Hymenoptera can initiate egg activation and embryo development. Eggs of the wasp Pimpla turionellae squeezed through a narrow capillary activate and develop into male larvae [36, 37]. Second, it was found that application of hydrostatic pressure or manual pulling of dorsal appendages iniated resumption of meiosis in Drosophila oocytes [38, 39]. It was further demonstrated that mechanical stimulation during ovulation triggered Drosophila egg activation via influx of calcium into the eggs [40]. Third, it was shown that unfertilized eggs of the frog Eleutherodactylus coqui can be easily activated by mechanical stimulation. It was proposed that spontaneous activation, which was observed in 34% of eggs, occurs in response to mechanical stress during oviposition [41]. Finally, Xenopus eggs can be activated by mechanical stimulation, such as pricking with a sharp needle. A wave of free cytosolic calcium is initiated starting from the point of prick activation [42, 43]. Of note, overactivation and activation were not discriminated in the abovementioned studies. Thus, based on the previous reports and the results obtained in the present study, we suggest that mechanical stress during oviposition may promote overactivation of Xenopus eggs”.

and

“Markedly, although spontaneous overactivation is a relatively rare phenomenon, which normally affects a small minority of ovulated eggs, our data demonstrate that mechanical or oxidative stress can greatly increase frequency of overactivation. In fact, egg overactivation becomes practically inevitable under high stress (Figure 4), [18]. Thus, it can be hypothesized that in some cases the proportion of overactivated eggs in natural egg populations may increase significantly, reflecting egg condition and stress intensity. Considering that mechanical stress accompanies oviposition in different species (see discussion above), the increased manifestation of egg overactivation under stress-inducing conditions requires additional investigation. The approaches that can prevent or attenuate overactivation should be pursued with the purpose of increasing egg quality. Considering close resemblance of regulatory mechanisms and high functional similarity of frog and mammalian eggs, the findings in frogs can possibly be extended to mammalian eggs with applications in assisted reproduction. In addition, studies of overactivated eggs can expand our understanding of cell death by disclosing alternative physiological mechanisms. Further studies are necessary to delineate in detail intracellular molecular events in overactivated eggs”.

Authors need to explain why they think that this research has any importance for the gamete and reproductive biology field.

We have narrowed and detailed the importance of our research for the gamete and reproductive biology. Now, the importance of this study is reflected in the second half of the newly introduced abovementioned paragraph (lines 453-462).

“Considering that mechanical stress accompanies oviposition in different species (see discussion above), the increased manifestation of egg overactivation under stress-inducing conditions requires additional investigation. The approaches that can prevent or attenuate overactivation should be pursued with the purpose of increasing egg quality. Considering close resemblance of regulatory mechanisms and high functional similarity of frog and mammalian eggs, the findings in frogs can possibly be extended to mammalian eggs with applications in assisted reproduction. In addition, studies of overactivated eggs can expand our understanding of cell death by disclosing alternative physiological mechanisms. Further studies are necessary to delineate in detail intracellular molecular events in overactivated eggs”.

Comments on the Quality of English Language

There are some grammatical and other errors for example: " Water-soluble Adult wild-type female frogs Xenopus laevis were purchased from 99 Shimizu (Kyoto, Japan) and maintained in dechlorinated water at the ambient 100 temperature of 21–23°C." The whole text should be checked.

As advised, the whole text has been checked carefully and errors corrected.

Round 2

Reviewer 1 Report (Previous Reviewer 2)

Comments and Suggestions for Authors

The legend to Fig.2 is misleading it does not accurately describes what is shown in each panel

In Fig.3 and 5 Western blots there are no loading controls

Comments on the Quality of English Language

The English is inadequate

Round 3

Reviewer 1 Report (Previous Reviewer 2)

Comments and Suggestions for Authors

manuscript was improved

Comments on the Quality of English Language

ok, minor editing

Author Response

Thank you for your kind suggestion regarding the refinement of our English.

As indicated in the PDF file, we have made several corrections based on the professional advice.

We hope that our revisions will be satisfactory for acceptance by IJMS.

This manuscript is a resubmission of an earlier submission. The following is a list of the peer review reports and author responses from that submission.

Round 1

Reviewer 1 Report

Comments and Suggestions for Authors

The authors mention that spontaneous overactivation occurred quite rarely in Xenopus eggs (a number of around 1% is mentioned). For the graphs (eg Figure 1B, C, D) it would be informative to present the numbers of eggs that these graphs are based on. The figure legend mentions that ‘more than 5 cells were analyzed’, which can also be 100. Presumably, the numbers are around 5 per group? How many groups per time point?

Another question regarding the low percentage of eggs that spontaneously overactivate is the significance of the findings described in this manuscript. Presumably, large numbers of eggs are produced so that a good number survives to be fertilized. It seems logical that there is a certain level of redundancy. It needs to be better described why the findings in this manuscript are relevant.

Similarly to the previous remark, what was the percentage of eggs that underwent apoptosis, as presented in Figure 2. For the data points in 2B, C, D it is mentioned that ‘More than five cells of each morphological type were analyzed in panels B, C and D’. Are then biologically independent groups combined to obtain standard deviations?

Line 173: ‘whether spontaneous overactivation in the absence of any activating stimuli’ is a bit confusing, as one would assume that ‘spontaneous’ is by definition in the absence of activating stimuli.

Statistics are lacking in Figure 3B and 3D.

Figure 1 C and 2C are both demonstrating ATP content. Why is the presentation (Au in 1 and cpm in 2) different? In Figure 4D the luminescence seems to be a totally different scale?

Statistics are lacking in Figure 5B and 5D.

Statistics are lacking in Figure 6 A and 6B.

Figure 3A and Figure 5A: Using a ‘nonspecific protein’ as a normalization factor is not acceptable. It is not known whether this protein is expressed at similar levels in the cells at different conditions. A proper antibody that detects a known protein should be used. Moreover, the bands on the immunoblot were ‘detected by color development catalyzed by peroxidase in the presence of hydrogen peroxide and diaminobenzidine tetrahydrochloride.’. This does not seem to be a linear process, so it can be argues whether the intensity of the bands can be used to quantify the expression levels of the proteins.

For me it is not clear what is meant with extracellular compartments of eggs. What is the extracellular compartment of eggs? Please explain.

In the Introduction it is mentioned that egg overactivation is rare on is only observed in several percentages of cells. In the beginning of the Discussion it is stated however that ‘Egg overactivation is an abnormal physiological process that often accompanies ovulation and spawning in frogs’ Thse statements appear to contradict.

It is concluded that mechanical stress during egg laying causes the spontaneous overactivation. This does not explain however the observation that only 1% of the spawned eggs undergoes overactivation. Indeed the authors demonstrate that mechanical stress can give rise to overactivation, but they do not demonstrate that spawning gives rise to mechanical stress.

The final remark in the conclusions: ‘These studies will have clear and significant implications for reproductive biology’.  Please give examples of these implications. It seems that overactivation in a small percentage of eggs, is just a natural phenomenon a kind of competition to assure that only the best eeggs are fertilized.

Minor comments:

Line 99 adult instead of Adult.

In the M&M both hours and hrs are used. Please keep it consistent.

Line 185 ‘in late in apoptosis’ seems not correct, change to ‘in late apoptosis’.

Line 269 ‘(‘ is lacking in front of ‘Figure’.

Reviewer 2 Report

Comments and Suggestions for Authors

There are some grammatical and other errors for example: " Water-soluble Adult wild-type female frogs Xenopus laevis were purchased from 99 Shimizu (Kyoto, Japan) and maintained in dechlorinated water at the ambient 100 temperature of 21–23°C." The whole text should be checked.

It is very unclear to me why the authors studied the spontaneous egg over activation if as they admitted: "Normally, the proportion of overactivated eggs is quite low in natural egg  populations; it rarely exceeds several percent" .  What is the overall meaning of their findings? Also when you damage the eggs by vigorous pipetting no wander that these eggs die, I would not call it a mechanical stress comparable to the possible mechanical stress in the genital tract.

Authors need to explain why they think that this research has any importance for the gamete and reproductive biology field.

Comments on the Quality of English Language

There are some grammatical and other errors for example: " Water-soluble Adult wild-type female frogs Xenopus laevis were purchased from 99 Shimizu (Kyoto, Japan) and maintained in dechlorinated water at the ambient 100 temperature of 21–23°C." The whole text should be checked.